# Regional Evaluation of Two SARS-CoV-2 Antigen Rapid Diagnostic Tests in East Africa

Muna Affara,[a,b] Hakim Idris Lagu,[b] Emmanuel Achol,[b] Neema Omari,[a,b] Grace Ochido,[a,b] Eric Kezakarayagwa,[c] Francine Kabatesi,[c] Cassien Nduwimana,[c] Anatole Nkeshimana,[c] Donald Duku Samson,[d] Gwokpan Awin Nykwec,[d] Joseph Daniel Wani Lako,[d] Michael Lasuba,[d] Lul Lojok Deng,[d] Maria Ezekiely Kelly,[e,f] Peter Bernard Mtesigwa Mkama,[e,f] Alex Magesa,[e,f] Salum Said Ali,[g] Sabra Amour Rashid,[g] Godfrey Pimundu,[h] Tonny Muyigi,[h] Susan Ndidde Nabadda,[h] Robert Rutayisire,[i] Alice Kabanda,[i] Emmanuel Kabalisa,[i] Jürgen May,[a,j,k] Eric Nzeyimana,[b] Michael Katende,[b] Florian Gehre[a,b]

aDepartment for Infectious Disease Epidemiology, Bernhard-Nocht-Institute for Tropical Medicine, Hamburg, Germany

bEast African Community (EAC), Arusha, Tanzania

cNational Institute of Public Health, Ministry of Health and Fight Against AIDS, Bujumbura, Burundi

dPublic Health Laboratory and National Blood Transfusion Centre, Ministry of Health, Juba, South Sudan

eMinistry of Health, Dodoma, Tanzania

fNational Public Health Laboratory, Dar es Salaam, Tanzania

gZanzibar National Public Health Laboratory, Stonetown, Zanzibar

hNational Health Laboratory and Diagnostic Services (NHLDS), Ministry of Health, Kampala, Uganda

iNational Reference Laboratory, Kigali, Rwanda

jGerman Center for Infection Research (DZIF), partner site Hamburg-Lübeck-Borstel-Riems, Hamburg, Germany

kTropical Medicine II, University Medical Center Hamburg-Eppendorf (UKE), Hamburg, Germany

Muna Affara and Florian Gehre contributed equally to this work and share first/last authorship. The order was determined by the corresponding author after negotiation.

**ABSTRACT** The clinical performance of two rapid antigen tests for the diagnosis of Severe Acute Respiratory Coronavirus (SARS-CoV-2) were regionally evaluated in East African populations. Swabs were collected from 1,432 individuals from five Partner States of the East African Community (Tanzania, Uganda, Burundi, Rwanda and South Sudan). The two rapid antigen tests (Bionote NowCheck COVID-19 Ag and SD Biosensor STANDARD Q COVID-19 Ag) were evaluated against the detection of SARS-CoV-2 RNA by the Reverse Transcription PCR (RT-PCR) gold standard. Of the concordant results with both RT-PCR and rapid antigen test data (862 for Bionote and 852 for SD Biosensor), overall clinical sensitivity was 60% and 50% for the Bionote NowCheck and the SD Biosensor STANDARD Q, respectively. Stratification by viral load, including samples with RT-PCR cycle thresholds (Ct) of <25, improved sensitivity to 90% for both rapid diagnostic tests (RDTs). Overall specificity was good at 99% for both antigen tests. Taken together, the clinical performance of both Ag-RDTs in real world settings within the East African target population was lower than has been reported elsewhere and below the acceptable levels for sensitivity of >80%, as defined by the WHO. Therefore, the rapid antigen test alone should not be used for diagnosis but could be used as part of an algorithm to identify potentially infectious individuals with high viral load.

**IMPORTANCE** Accurate diagnostic tests are essential to both support the management and containment of outbreaks, as well as inform appropriate patient care. In the case of the SARS-CoV-2 pandemic, antigen Rapid Diagnostic Tests (Ag-RDTs) played a major role in this function, enabling widespread testing by untrained individuals, both at home and within health facilities. In East Africa, a number of SARS-CoV-2 Ag-RDTs are available; however, there remains little information on their true test performance within the region, in the hands of the health workers routinely carrying out SARS-CoV-2 diagnostics. This study contributes test performance data for two commonly used SARS-CoV-2 Ag-RDTs in East Africa, which will help inform the use of these RDTs within the region.

Address correspondence to Muna Affara, affara@bnitm.de.

The authors declare no conflict of interest.

**KEYWORDS** SARS-CoV-2, EAC mobile laboratory, capacity building, East African community, outbreak response, antigen rapid diagnostic test, RT-PCR

All seven Partner States of the East African Community (EAC) were affected by the global SARS-CoV-2 pandemic in 2020 (Burundi, Rwanda, Democratic Republic of Congo, Tanzania, Kenya, Uganda, and South Sudan). Although differing national control measures in the countries were applied (ranging from restrictions of free population movement, partial curfews, to complete lockdowns), the requirement for negative PCR was commonplace to enter or exit the countries across land and air borders. The PCR test results were captured in the "EAC Pass app" (1) to facilitate the movement of commuters, tourists and goods in the region. By 2022, the majority of countries stopped mandatory testing, with valid SARS-CoV-2 vaccination certificates accepted in place of negative tests. However, given the low vaccination coverage across the East African Partner States (2), SARS-CoV-2 testing remains a requirement for the majority of the unvaccinated population crossing borders. Given this continued need for testing and the unpredictable emergence of future, novel SARS-CoV-2 variants, it is conceivable that mandatory testing requirements for all people at border crossings within the EAC may be reinstalled. For this scenario the widespread regional use of rapid diagnostic tests (RDTs) would have several advantages, as they require a lower operator skill set, have a rapid diagnostic turn-around-time (15 min), do not need expensive laboratory equipment and are therefore cost-effective. If they were also capable of identifying a critical proportion of infectious individuals in East Africa, they would be an acceptable alternative to PCR tests. Many SARS-CoV-2 antigen RDTs have been evaluated globally (3–5), including through the framework of the Foundation of Innovative Diagnostics (FIND) (6), but there is very little performance information in East African populations, especially in a real world setting, i.e. in the hands of the laboratory personnel carrying out the testing of local populations at points-of-entry, community-level, or health facilities (7–9).

Since 2017, the EAC has been operating a mobile laboratory network, together with the six National Public Health Laboratories (NPHLs) in East Africa (DRC only joined the EAC in 2022), with the aim to harmonize laboratory procedures for diagnostics of viral (hemorrhagic) infections in the region. Since the emergence of SARS-CoV-2, this laboratory network has significantly contributed to the national and regional pandemic SARS-CoV-2 response (10, 11).

For the present study, we were utilizing the established network to validate the performance of two SARS-CoV-2 RDTs at deployed mobile laboratories (South Sudan, Zanzibar/Tanzania) and centralized NPHLs (Burundi, Rwanda, Uganda), respectively. We evaluated in detail, the sensitivity and specificity of the RDTs in comparison to gold standard qRT-PCR in the five countries. This RDT field validation was carried out between 2020 and 2021 by the trained front-line laboratory personnel that routinely performed SARS-CoV-2 testing in their respective country and represents the first such regional evaluation.

## RESULTS

**Patient enrollment and demographics of study participants.** Between 2020 and 2021, 1,432 individuals participated in the regional study, across the five EAC Partner States and samples were collected (see Table 1 and flowchart Fig. 1).

Results for both the reference molecular RT-PCR test and the Ag-RDT were available for 862 participants in the Bionote NowCheck COVID19 Ag Test sampling group and 852 participants in the SD Biosensor STANDARD Q COVID-19 Ag Test sampling group. The demographic characteristics were similar across the five EAC Partner States for mean age, ranging from 30 to 42 years (Table 1). Gender was balanced in Rwanda and Uganda (51% and 53% female, respectively); however, in Zanzibar females were over-represented at 63% and in Burundi and South Sudan there was an underrepresentation at 14% and 45%, respectively. History of recent travel was low in all study populations, except for South Sudan, where 47% of participants had a history of recent travel, due

**TABLE 1** Demographics of study population in five EAC Partner states

| Characteristics | Zanzibar | | | Burundi | | | South Sudan | | | Rwanda | | | Uganda | | |
|---|---|---|---|---|---|---|---|---|---|---|---|---|---|---|---|
| | Overall | Sub-analysis Bionote | Sub-analysis SD Biosensor | Overall | Sub-analysis Bionote | Sub-analysis SD Biosensor | Overall | Sub-analysis Bionote | Sub-analysis SD Biosensor | Overall | Sub-analysis Bionote | Sub-analysis SD Biosensor | Overall | Sub-analysis Bionote | Sub-analysis SD Biosensor |
| Total (N) | 275 | 171 | 171 | 262 | 82 | 82 | 223 | 102 | 94 | 327 | 162 | 160 | 345 | 345 | 345 |
| Age (mean [min-max], yrs) | 42 (1-92) | 45 (1-90) | 45 (1-90) | 39 (5-65) | 37 (19-65) | 37 (19-65) | 30 (10-69) | 28 (14-68) | 33 (13-67) | 41 (18-63) | 40 (18-63) | 37 (13-86) | 41 (2-87) | 41 (2-87) | 41 (2-87) |
| Gender (% Females, [n/N]) | 63% (174/275) | 68% (110/162) | 64% (110/171) | 14% (36/262) | 33% (27/82) | 33% (27/82) | 45% (101/223) | 40% (41/102) | 46% (43/94) | 51% (166/327) | 55% (89/162) | 52% (82/159) | 53% (184/345) | 54% (186/342) | 54% (186/342) |
| Recent travel (% Yes, [n/N]) | 9% (22/240) | 12% (20/162) | 11% (19/167) | 5% (12/262) | 13% (11/82) | 13 (11/82) | 47% (104/223) | 30% (31/102) | 56% (53/94) | 0% (0/327) | 0% (0/162) | 0% (0/160) | 0.3% (1/345) | 0.3% (1/338) | 1% (2/338) |
| Symptoms present (% Yes, [n/N]) | 94% (259/275) | 97% (166/171) | 89% (152/171) | 100% (262/262) | 100% (82/82) | 85% (70/82) | 8% (18/223) | 8% (8/102) | 22% (21/94) | 4% (14/327) | 8% (12/160) | 0% (0/160) | 80% (279/345) | 81% (279/345) | 82% (282/345) |
| Hospitalized (n, % Yes) | 47% (129/275) | 58% (98/169) | 52% (88/169) | 8% (12/262) | 24% (20/82) | 24% (20/82) | 2% (4/223) | 2% (2/102) | 9% (8/94) | 0% (0/327) | 0% (0/162) | 0% (0/160) | 14% (51/345) | 17% (51/293) | 17% (57/344) |
| Days from symptom onset (median [Q1-Q3]; N) | 7 (3-7), 275 | 7 (3-7), 170 | 7 (3-7), 170 | 6 (3-9), 262 | 7 (4-9), 82 | 7 (4-7), 82 | 7% (13/197) | 5% (5/102) | 5% (5/94) | 8% (27/327) | 15% (25/162) | 3% (5/160) | 3 (1-14), 94 | 3 (1-14), 94 | 3 (1-3), 94 |
| Seegene PCR Positivity (%, [n/N]) | 77% (132/171) | 77% (132/171) | 77% (132/171) | 38% (31/82) | 38% (31/82) | 38% (31/82) | 5% (5/110) | 4% (4/102) | 80% (4/5) | 14% (24/163) | 14% (23/162) | 3% (4/160) | 27% (94/345) | 27% (94/345) | 27% (94/345) |
| Bionote NowCheck COVID-19 Ag Test Positivity (%, [n/N]) | 40% (68/171) | 40% (68/171) | 40% (68/171) | 3% (9/262) | 9% (7/82) | 9% (7/82) | 5% (5/111) | 4% (4/102) | 4% (4/94) | 2% (4/263) | 0% (0/162) | 3% (4/160) | 24% (85/345) | 24% (85/345) | 24% (85/345) |
| SD Biosensor STANDARD Q COVID-19 Ag Test Positivity (%, [n/N]) | 40% (69/171) | 40% (69/171) | 40% (69/171) | 4% (11/262) | 6% (5/82) | 6% (5/82) | | 4% (4/102) | 4% (4/94) | | 0% (0/162) | 3% (4/160) | 24% (85/345) | 24% (83/345) | 25% (85/345) |

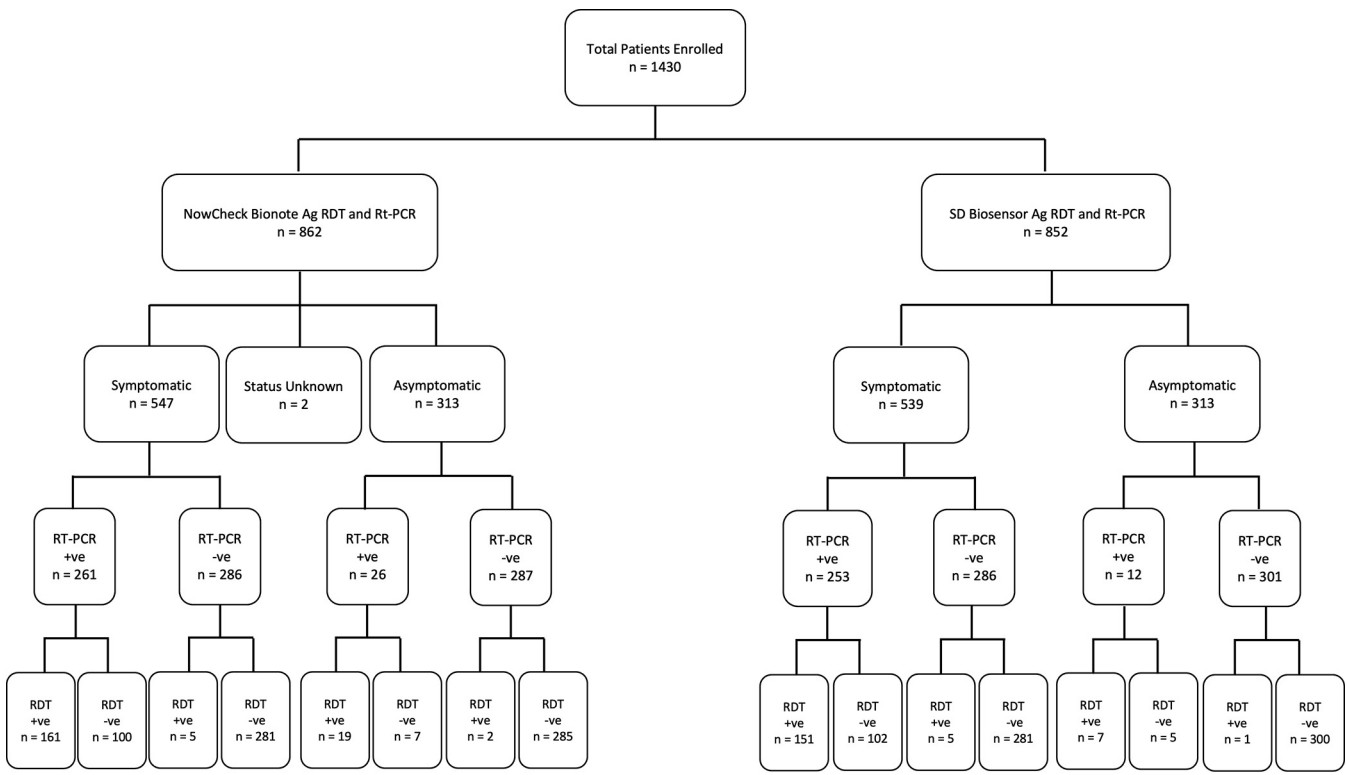

**FIG 1** Flow diagram of sample recruitment and diagnostic analysis in the 5 EAC Partner States.

to the location of the mobile laboratory at the South Sudan/Uganda border. In Zanzibar, Burundi and Uganda, the majority of participants were symptomatic at 94%, 100%, and 80%, respectively, reflecting the hospital-based recruitment. This was lower in the community-based recruitment at 8% in South Sudan and 4% in Rwanda. Hospitalization of participants was low in all study sites, except Zanzibar where almost half of those recruited in the study were hospitalized (47%). Days from symptom onset was captured in Zanzibar, Burundi and Uganda, median range 3 to 7 days (for a detailed overview see Table 1). Sub-analysis of the Bionote NowCheck COVID19 Ag Test sampling group and the SD Biosensor STANDARD Q COVID-19 Ag Test sampling group was additionally carried out to ensure that the characteristics of the sub-groups were not skewed (Table 1). Overall, age and gender distributions were similar across all groups. A slightly lower proportion of symptomatic patients was observed in the Burundi SD Biosensor subgroup compared to the overall (85% versus 100%) and a slightly higher proportion in the South Sudan SD Biosensor subgroup (22% versus 8%). Increased numbers of hospitalized patients were also observed in the Burundi and South Sudan sub-groups, relative to the overall (see Table 1 for detailed overview). PCR and rapid test positivity rates were maintained across all the groups.

**Performance of the two RDTs in comparison to RT-PCR gold standard.** In total we correlated RT-PCR results for 862 samples with the outcome of the Bionote NowCheck COVID-10 Ag test, and 852 samples with the outcome of the SD Biosensor STANDARD Q COVID-19 Ag Test. For an overview see the 2 × 2 contingency tables below (Table 2).

While the Bionote NowCheck COVID-19 Ag test showed a nonsignificant, higher sensitivity than the SD Biosensor STANDARD Q COVID-19 Ag (60% versus 53%) in its overall performance (Table 3), neither of the tests reached satisfactory clinical sensitivity levels (>80% as per WHO guidelines). In the case of both RDTs, test sensitivity was lower than that reported by the manufacturer (94% for SD Biosensor STANDARD Q COVID-19 Ag and 96% for Bionote NowCheck COVID-19 Ag test). Stratification of qRT-PCR ct values, to only include those samples with higher viral load, improved sensitivity to 90% for both RDTs at Ct < 25 (see Table 3). As indicated by the overlapping 95%CI,

**TABLE 2** Bionote NowCheck COVID-19 Ag and SD Biosensor STANDARD Q COVID-19 Ag Test performance in relation to the gold standard SeeGene qRT-PCR

| | Seegene | | | | |
|---|---|---|---|---|---|
| | | Seegene RT-PCR −ve | | Seegene RT-PCR +ve | |
| Rapid diagnostic test | | n = 575 | | n = 287 | |
| Nowcheck Bionote Ag RDT | N = 862 | −ve | +ve | −ve | +ve |
| | | 568 | 7 | 107 | 180 |
| | | 99% | 1% | 37% | 63% |
| | Seegene | | | | |
| | | Seegene RT-PCR −ve | | Seegene RT-PCR +ve | |
| | | n = 587 | | n = 265 | |
| SD Biosensor Ag RDT | N = 852 | −ve | +ve | −ve | +ve |
| | | 581 | 6 | 107 | 158 |
| | | 99% | 1% | 40% | 60% |

none of the tests outperformed the other. Both tests showed comparable high clinical specificity at 99% (98% to 100%), in line with the manufacturers report (see Table 3).

To further understand the lower sensitivity of rapid tests we analyzed concordant (RT-PCR positive/RDT positive) and discordant (RT-PCR positive/RDT negative) test results, in particular in relation to the Ct values as proxy for viral loads (Fig. 2).

For the concordant positive samples of the Bionote NowCheck Ag RDT (n = 180), the median Ct was 27 (Q1-Q3 21 to 32), whereas discordant, false-negative RDTs (n = 107) had a higher median Ct of 35 (Q1-Q3 32 to 37). For the SD Biosensor STANDARD Q concordant cases (n = 158) had a median Ct of 26 (range 21–33) while discordant cases (n = 107) showed a higher Ct of 35 (range 31 to 37). In both instances, lower Ct values, which correlate with higher viral loads, increased the sensitivity of the RDTs.

**The impact of clinical patient status on RT-PCR and RDT outcome.** We wanted to investigate whether the presence of symptoms in COVID-19 suspects was a predictor of test outcome.

We stratified the days to symptoms onset and related it to test outcome (see Table 4). We had data from Zanzibar, Burundi, and Uganda for a total of n = 558 patients that had symptoms for a median of 6 days (Q1 to Q3, 3 to 8 days) prior to enrollment in the study. When further looking at the test outcome through the course of the acute infection (0 to 3 days; 4 to 7 days; >8 days) we found the highest proportion (36%) of RT-PCR test positives was found when the symptom onset was 4 to 7 days prior. Similarly, the two rapid tests had the highest proportions of positive cases in the same 4 to 7 days window with 26% for both the Bionote NowCheck Ag RDT and the SD Biosensor STANDARD Q (see Table 4).

Furthermore, we tested the performance of both test kits relative to the RT-PCR in asymptomatic and symptomatic patients, by analyzing the positive predictive values (PPV). We observed a higher PPV in the symptomatic group compared to the asymptomatic group for both RDTs; PPV symptomatic (95% CI): 97% (93 to 99%), relative to the PPV asymptomatic (95% CI): 90% (70 to 97%) and 87% (48 to 98%) for Bionote NowCheck and SD Biosensor, respectively (Table 5).

Stratification of all RT-PCR positive cases by symptoms, revealed an insignificant difference in PCR Ct values between symptomatic and asymptomatic participants (Fig. 3),

**TABLE 3** Sensitivity and specificity of the RDTs compared to qRT-PCR

| Test performance | Bionote NowCheck COVID-19 Ag test | SD biosensor STANDARD Q COVID-19 Ag |
|---|---|---|
| Overall Sensitivity (95% CI), N | 60% (55 - 66), 297 | 53% (49 - 56), 297 |
| Sensitivity Ct ≤ 33, (95% CI), N | 77% (70 - 83), 163 | 72% (65 - 79), 163 |
| Sensitivity Ct ≤ 25, (95% CI), N | 90% (86 - 99), 78 | 90% (84 - 98), 78 |
| Overall Specificity (95% CI), N | 99% (98 - 100), 575 | 99% (98 - 100), 587 |

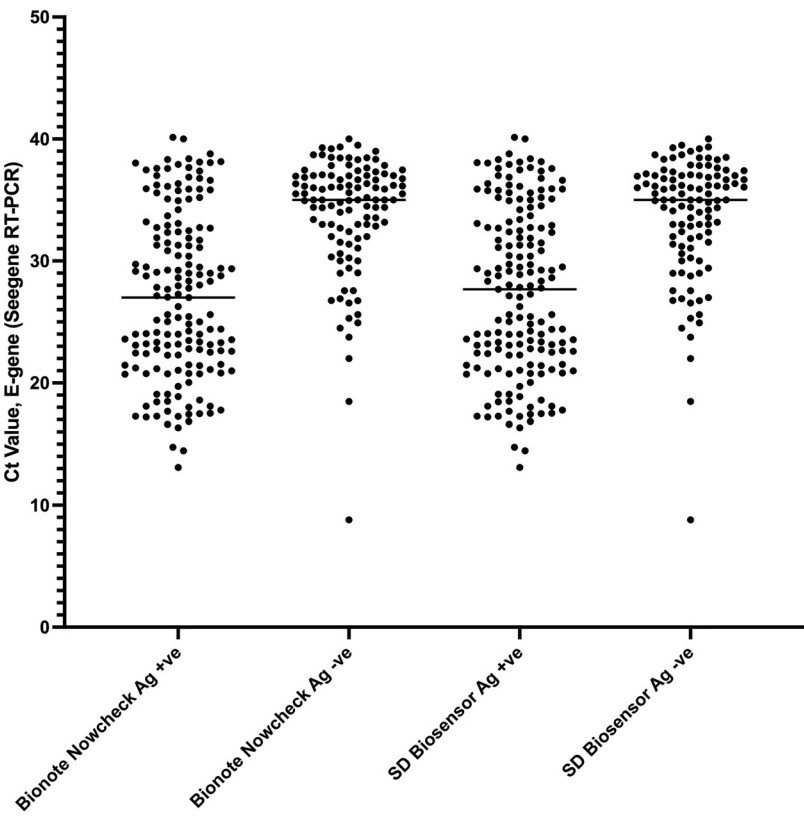

**FIG 2** Cycle Threshold (Ct) values for SARS-CoV-2 E-gene in Bionote NowCheck and SD Biosensor rapid diagnostic test positive and negative patients. Median Ct Bionote NowCheck Ag RDT positive 27, and negative 35. Median Ct SD Biosensor Standard Q Ag RDT positive 25.6 and negative 35.

as the Median Ct (and range) for symptomatic cases was 29 (24 to 36) and for asymptomatic 31.5 (30 to 35), respectively.

## DISCUSSION

Here, we present the first regional evaluation study for SARS-CoV-2 RDTs in East Africa, conducted at the height of the COVID-19 pandemic in the region between 2020 and 2021. We evaluated the tests in five countries in collaboration with the NPHLs, which are mandated to respond to outbreaks. In contrast to test evaluation in research laboratories, the performance of the RDTs at the hands of front-line laboratory workers in real field settings, in hospitals and health facilities in East Africa, is most relevant.

We selected the Seegene qRT-PCR as the gold standard, as it showed satisfactory performance (limit of detection [LOD]: 50 copies per reaction, specificity: 100%), as evaluated by the Foundation for Innovative New Diagnostics (FIND) (12). Antigen RDT test selection was based on (1) shortlisting as part of the FIND evaluation and (2) availability and use in the East African region at the time of study. Although we originally attempted to procure four RDTs, due to the limitations in air travel, shipping and availability in East Africa during the onset of the pandemic in 2020, we were only able to validate two RDTs.

The overall sensitivity of the RDTs to detect low viral load patients was much lower than both that reported by the manufacturers and relative to the gold qRT-PCR standard.

**TABLE 4** Test sensitivity in relation to onset of symptoms

| Onset of symptoms | n | Seegene RT-PCR | Bionote NowCheck | SD Biosensor STANDARD Q |
|---|---|---|---|---|
| Days < 0–3 (*n*, %) | 191 | 54 (28%) | 43 (22%) | 36 (18%) |
| Days 4–7 (*n*, %) | 226 | 82 (36%) | 60 (26%) | 59 (26%) |
| Days >8 (*n*, %) | 141 | 40 (28%) | 18 (12%) | 19 (13%) |

**TABLE 5** Test performance in symptomatic and asymptomatic COVID-19 suspects (positive predictive values)

| Test performance | Bionote NowCheck COVID-19 Ag test | SD Biosensor STANDARD Q COVID-19 Ag |
|---|---|---|
| Symptomatic PPV (95% CI), N | 97% (93–99%), 547 | 97% (93–99%), 539 |
| Asymptomatic PPV (95% CI), N | 90% (70–97%), 313 | 87% (48–98%), 313 |

This may be attributed to a number of factors, including the possibility of skewed selection of higher viral load samples for manufacturer test validation (9). Therefore, RDTs might give false-negative results during large-scale, cross-sectional screening of populations (such as truck-drivers, tourists, commuters). This finding is consistent with other recently published studies in Africa (8, 9), and are especially relevant to mobile laboratory settings such as the EAC Mobile laboratory in Nimule/South Sudan screening transient populations, which was reflected in a high percentage of recent travel history and low percentage of symptomatic cases. As a matter of fact, seven EAC Mobile Laboratories were deployed to respective border regions during the pandemic (11), screening over 1 million people with RT-PCR, with the majority being asymptomatic. Therefore, the application of RDT screening in this context, should consider a high level of negative results, which will need to be further interpreted based on clinical and epidemiological information, and possible retesting by RT-PCR.

In samples with high viral loads, however, the sensitivity of both tests increased (to 75% to 90%), consistent with other studies (13, 14). Therefore, the evaluated Ag RDTs might still be useful to identify most SARS-CoV-2 infections with significant transmission risk, where they could play an important role to rapidly isolate infectious suspects at points-of-entry, such as truck drivers at border crossings. Despite our study showed higher PPVs in symptomatic patients 97% (CI95:93 to 99), relative to asymptomatic patients (90% [CI95:70 to 97] and 87% [CI95:48 to 98] for Bionote NowCheck and SD Biosensor, respectively), there still appears to be a high possibility of false-negative RDT

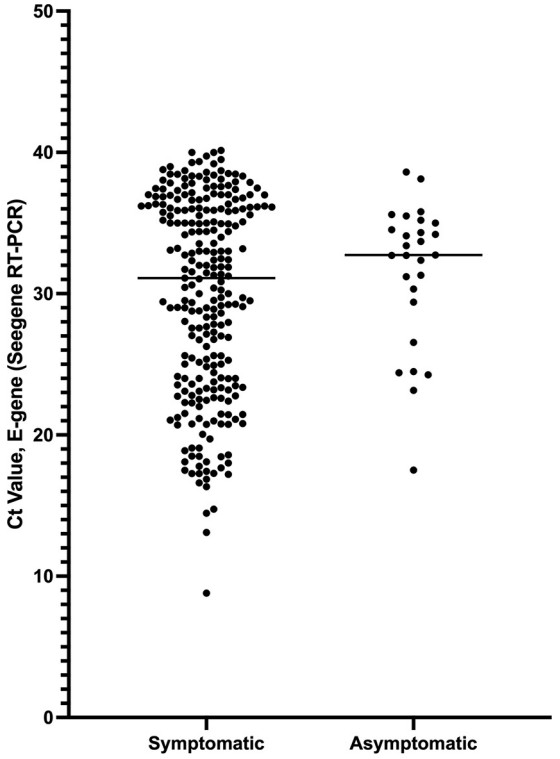

**FIG 3** Cycle Threshold (Ct) values for SARS-CoV-2 E-gene in symptomatic and asymptomatic patients. Median Ct symptomatic 29 (24–36) and asymptomatic 31.5 (30–35).

results, even in symptomatic patients. Therefore, in line with The World Health Organization (WHO) recommendations, further RT-PCR testing should still be recommended in East Africa for this symptomatic, RDT negative patient group, particularly given the low reported SARS-CoV-2 prevalence rates in the EAC Partner States (2). Our sensitivity of 53% (CI95: 49 - 56) on the Biosensor STANDARD Q are comparable with findings from an independent Ugandan study, which also documented a low sensitivity of 70.0% (CI95: 60 to 79) that increases with higher viral loads (i.e., lower Ct values) (7). Similar to our findings, the Ugandan authors concluded that the Biosensor STANDARD Q RDT has a "less than optimal performance," as the WHO recommends a sensitivity of >80% for SARS-CoV-2 antigen RDTs (15). This also applies to the performance of the Bionote NowCheck evaluated in our setting. Both rapid antigen tests demonstrated high specificity (99% [98 to 100]), suggesting that positive Ag RDT results do not require further RT-PCR confirmation. This could prove beneficial in the context of practical implementation for practicing professions, working for example at points of entry, where an algorithm can be established to reduce diagnostic turn around time (TAT) and minimize the use of expensive and time-consuming molecular tests.

A major limitation of the study was that we did not have further information of viral strains or variants, as SARS-CoV-2 whole-genome sequencing was not yet readily available for NPHLs at the time of the study. Recent publications from a number of East African Partner States, including Tanzania, Kenya, Uganda, Rwanda, Burundi, and South Sudan reveal that global variants of concern (VOCs), including the Beta, Delta, and Eta variants, were circulating at the time of this study (16–18). In addition, a number of regionally specific lineages appear to have evolved in East Africa, which show very low prevalence elsewhere in the world (16, 18, 19). Given that rapid antigen tests, in contrast to molecular tests, only target a singular, genetically conserved nucleocapsid viral protein, there is a chance that strain variation could explain the low sensitivity observed in this study, and that these tests might work differently or even better in more recently emerged variants. A number of recent studies have demonstrated an impact of SARS-CoV-2 variants on rapid antigen test sensitivity (20–22). However, other publications suggest that there is no significant difference (23).

Overall, our study demonstrates that in real world field settings, with potential suboptimal conditions and sampling/operator variance, the two evaluated RDTs demonstrate low sensitivity overall in the target group population in East Africa. However, there may be a role in the rapid confirmation of SARS-CoV-2 infection in symptomatic patients presenting with high viral loads, which may represent the most infectious individuals, in particular between 4 and 7 days after symptoms onset, a phase of high transmission risk.

## MATERIALS AND METHODS

**Study sites.** The study was carried out in five EAC Partner States, through the respective National Public Health Laboratories (NPHLs), namely Burundi (Institut National de Santé Publique), Rwanda (National Reference Laboratory), South Sudan (Public Health Laboratory and National Blood Transfusion Services), Uganda (Central Public Health Laboratories). The Zanzibar National Public Health Laboratory received EAC mobile laboratory support from the National Health Laboratory (Tanzania). Recruitment was conducted in different health settings in the various EAC Partner States. Hospital based recruitment was performed in Burundi (at the Clinique Prince Louis Rwagasore and the Bon Accueil, Kamenge, Kanyosha, Bubanza field sites); Uganda (at the regional referral hospitals in Arua, Mbale, Soroti); and in Zanzibar (at the Mnazi Mmoja Referral Hospital and Kwa Mtipura Primary Health Unit), respectively. In South Sudan, recruitment was conducted in the EAC mobile laboratory located in Nimule, a border town with Uganda, predominantly testing travelers entering South Sudan. In Rwanda, recruitment was both hospital-based (from Kibungo Referral Hospital and Kirehe District Hospital) and also during community-based mass screening events in the Eastern Provinces of the country.

**Clinical specimens.** A total of 1,432 respiratory samples (nasopharyngeal and throat swabs) were collected from suspect COVID-19 cases, taking all necessary biosafety precautions. Two or three swabs were taken per participant, dependent on participant acceptance. One swab was collected in appropriate transport media, e.g., Universal Viral Transport media, VTM (Copan Diagnostics Inc., Brescia, Italy) or DNA/RNA shield (Zymo Research, USA) for downstream RT-PCR diagnostics, and one swab in the proprietary storage solution for each of the two SARS-CoV-2 antigen RDTs under evaluation. Where only two swabs were collected, one swab was used for RT-PCR and the other swab with one of the RDTs. In these cases, selection was done to ensure that sampling was balanced between the two RDTs under evaluation. An outline of the testing algorithm is shown in Fig. 1.

**Viral RNA extraction and SARS-CoV-2 detection by RT-PCR.** RNA was extracted using the Zymo Research Quick RNA Viral kit (Zymo Research, USA), in accordance with the manufacturer's protocol. For

all extractions, 10μL Seegene internal control was added to each 400μL clinical specimen, for the downstream RT-PCR analysis. Extracted samples were eluted in 15–30μL elution buffer and stored at −80°C. Molecular testing as a reference gold-standard method was carried out using the Seegene Allplex 2019-nCoV Assay (Seegene, South Korea), according to the manufacturer's protocol.

**SARS-CoV-2 antigen tests.** All swab samples were tested with the two RDT under investigation: (i) SD Biosensor STANDARD Q COVID-19 Ag Test (24), and (ii) Bionote NowCheck COVID-19 Ag Test (25). Both RDTs were performed according to the respective manufacturer's instructions, and as soon as possible after specimen collection. All necessary reagents to perform the assays were provided by the manufacturer and no assay specific, specialized equipment was needed.

**Statistical analysis.** Statistical analysis was carried out using STATA/SE 14.2 and PRISM version 9. The sensitivity was calculated as the proportion of true positive results detected by the SD Biosensor STANDARD Q COVID-19 Ag test or Bionote NowCheck COVID-19 Ag Test compared to all positives by the quantitative reference PCR (Seegene Allplex 2019-nCoV Assay), expressed as a percentage. The specificity was calculated as the number of true negative specimens detected by the Rapid Diagnostic Test (RDT), among all negatives identified by the reference PCR, expressed as a percentage. The 95% confidence intervals were calculated to assess the level of uncertainty introduced by sample size.

**Ethics.** The study was approved in each participating EAC Partner State, through the relevant authorities: Burundi National Ethics Committee, Rwanda National Ethics Committee, South Sudan National Health Research Ethics Review, Uganda National Health Laboratory Services REC, Uganda National Council for Science and Technology and Zanzibar Medical Research and Ethics Committee (ZAMREC).

## ACKNOWLEDGMENTS

This study was funded by KfW Entwicklungsbank (KfW Development Bank) under grant reference BMZ 201568229. The funders had no role in study design, data collection or interpretation, or the decision to submit the work for publication.

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
