## [Reviewer comments · Microbiology Spectrum]

Microbiology Spectrum

Regional Evaluation of two SARS-CoV-2 Antigen Rapid Diagnostic Tests in East Africa

Muna Affara, Hakim Lagu, Emmanuel Achol, Neema Omari, Grace Ochido, Eric Kezakarayagwa, Francine kabatesi, Cassien Nduwimana, Anatole Nkeshimana, Donald Samson, Gwokpan Nykwec, Joseph Daniel Wani Lako, Michael Lasuba, Lul Deng, Maria Kelly, Peter Mkama, Alex Magesa, Salum Said Ali, Sabra Amour Rashid, Godfrey Pimundu, Tonny Muyigi, Susan Nabadda, Robert Rutayisire, Alice Kabanda, Emmanuel Kabalisa, Jürgen May, Eric Nzeyimana, Michael Katende, and Florian Gehre

Corresponding Author(s): Muna Affara, Bernhard-Nocht-Institut für Tropenmedizin

Review Timeline:

Submission Date:	November 29, 2022
Editorial Decision:	February 1, 2023
Revision Received:	February 15, 2023
Accepted:	February 24, 2023

Editor: Eleanor Powell

Reviewer(s): Disclosure of reviewer identity is with reference to reviewer comments included in decision letter(s). The following individuals involved in review of your submission have agreed to reveal their identity: Mehak Ali (Reviewer #1)

Transaction Report:

DOI: <https://doi.org/10.1128/spectrum.04895-22>

February 1, 2023

Dr. Muna Affara
Bernhard-Nocht-Institut für Tropenmedizin
Infectious Disease Epidemiology
Bernhard-Nocht-Straße 74
Hamburg 20359
Germany

Re: Spectrum04895-22 (Regional Evaluation of two SARS-CoV-2 Antigen Rapid Diagnostic Tests in East Africa)

Dear Dr. Muna Affara:

Thank you for submitting your manuscript to Microbiology Spectrum. As you will see your paper is very close to acceptance. Please modify the manuscript along the lines I have recommended. As these revisions are quite minor, I expect that you should be able to turn in the revised paper in less than 30 days, if not sooner. If your manuscript was reviewed, you will find the reviewers' comments below.

When submitting the revised version of your paper, please provide (1) point-by-point responses to the issues raised by the reviewers as file type "Response to Reviewers," not in your cover letter, and (2) a PDF file that indicates the changes from the original submission (by highlighting or underlining the changes) as file type "Marked Up Manuscript - For Review Only". Please use this link to submit your revised manuscript. Detailed instructions on submitting your revised paper are below.

Link Not Available

Sincerely,

Eleanor Powell

Reviewer comments:

Reviewer #1 (Comments for the Author):

Thank you for the privilege of reviewing your work.

The research work "Regional Evaluation of two SARS-CoV-2 Antigen Rapid Diagnostic Tests in East Africa" seems good; however, it needs a lot of improvement. Following comments may be considered for improvement:

1. The paper should be written as per guidelines of this journal, i.e. headings and sub headings like Materials and Methods, Results, Discussion and Conclusion should be there.
2. Only few studies are being cited. Literature may be further strengthened by adding more relevant studies from 2022 and 2023.
3. Results should be further elaborated with more scientific reasoning.
4. A separate brief section may be added for elaborating the practical implementation of the outcome of this research work to ease out practicing professionals.
5. You might be overstating the sensitivity of RDTs, which has been questioned.
6. The procedure for testing the RATs is a bit confusing. I'm envisioning something showing how many samples had a rapid antigen test done. And how both the kits were evaluated if only one sample was taken.

Reviewer #2 (Comments for the Author):

Thank you for your manuscript that describes an evaluation of RDT for SARS-COV-2 in East Africa. This is a critical study for the region.

I have only minor queries - do you know what variants were circulating at the time of testing and could you analyse differences in sensitivity and specificity according to variants?

You mention this in your discussion but we do not have that information for at least some of the countries.

Preparing Revision Guidelines

Please return the manuscript within 60 days; if you cannot complete the modification within this time period, please contact me. If you do not wish to modify the manuscript and prefer to submit it to another journal, please notify me of your decision immediately so that the manuscript may be formally withdrawn from consideration by Microbiology Spectrum.

Thank you for the privilege of reviewing "Regional Evaluation of two SARS-CoV-2 Antigen Rapid Diagnostic Tests in East Africa" (control no. Spectrum04895-22).

The research work Regional Evaluation of two SARS-CoV-2 Antigen Rapid Diagnostic Tests in East Africa" seems good; however, it needs a lot of improvement. Following comments may be considered for improvement:

1. The paper should be written as per guidelines of this journal, i.e. headings and sub headings like Materials and Methods, Results, Discussion and Conclusion should be there.
2. Only few studies are being cited. Literature may be further strengthened by adding more relevant studies from 2022 and 2023.
3. Results should be further elaborated with more scientific reasoning.
4. A separate brief section may be added for elaborating the practical implementation of the outcome of this research work to ease out practicing professionals.
5. You might be overstating the sensitivity of RDTs, which has been questioned.
6. The procedure for testing the RATs is a bit confusing. I'm envisioning something showing how many samples had a rapid antigen test done. And how both the kits were evaluated if only one sample was taken.

Response to reviews

We would like to thank the reviews for their comments on the manuscript, which have been beneficial in improving the paper. We have tried to address all comments raised by the reviewers and outline them below point by point.

We hope you find our responses and additions to minor comments raised on the manuscript satisfactory, which we have marked in the document by underlining. We hope that with these modifications, the manuscript will be accepted for publication in Microbiology Spectrum.

Yours sincerely,
Muna Affara

Reviewer 1:

1. The paper should be written as per guidelines of this journal, i.e. headings and sub headings like Materials and Methods, Results, Discussion and Conclusion should be there.

Response: We have ensured that headings and sub-headings are included in all sections.

2. Only few studies are being cited. Literature may be further strengthened by adding more relevant studies from 2022 and 2023.

Response: We acknowledge this point and have addressed this through the inclusion of more relevant literature, particularly from the East African region. We have included more recent studies from 2022 and 2023, including relevant studies on whole genome sequencing to understand the variants circulating in East Africa.

3. Results should be further elaborated with more scientific reasoning.

Response: We have elaborated further in the results and discussion sections on the scientific findings.

4. A separate brief section may be added for elaborating the practical implementation of the outcome of this research work to ease out practicing professionals.

Response: We have added a sentence in the discussion section to address this.

5. You might be overstating the sensitivity of RDTs, which has been questioned.

Response: We have clearly stated the poor sensitivity of these Ag RDTs under evaluation and the limitations of using RDTs, where you are likely to have a high false negative rate.

6. The procedure for testing the RATs is a bit confusing. I'm envisioning something showing how many samples had a rapid antigen test done. And how both the kits were evaluated if only one sample was taken.

Response: We acknowledge this comment and have modified the flow diagram in figure 1 to more clearly demonstrate the sample recruitment and testing work flow.

Reviewer 2:

1. I have only minor queries - do you know what variants were circulating at the time of testing and could you analyse differences in sensitivity and specificity according to variants?
You mention this in your discussion but we do now have that information for at least some of the countries.

Response: This is an interesting point and one we raised as a limitation in the manuscript. We have adjusted this section to include information from recent publications describing the variants circulating in the East African region at this time and the implications of this on Ag RDT sensitivity.

February 24, 2023

Dr. Muna Affara
Bernhard-Nocht-Institut für Tropenmedizin
Infectious Disease Epidemiology
Bernhard-Nocht-Straße 74
Hamburg 20359
Germany

Re: Spectrum04895-22R1 (Regional Evaluation of two SARS-CoV-2 Antigen Rapid Diagnostic Tests in East Africa)

Dear Dr. Muna Affara:

Your manuscript has been accepted, and I am forwarding it to the ASM Journals Department for publication. You will be notified when your proofs are ready to be viewed.

Sincerely,

Eleanor Powell
Editor, Microbiology Spectrum
